A new species of alien land flatworm in the Southern United States

Justine Jean-Lou justine@mnhn.fr 1
Gastineau Romain 2
Gey Delphine 3
Robinson David G. 4
Bertone Matthew A. 5
Winsor Leigh 6
1 ISYEB, Institut de Systématique, Évolution, Biodiversité, Museum national d’Histoire Naturelle , Paris , France
2 University of Szczecin , Szczecin , Poland
3 MCAM Molécules de Communication et Adaptation des Microorganismes, Museum national d’Histoire Naturelle , Paris , France
4 Academy of Natural Sciences of Drexel University , Philadelphia , PA , United States of America
5 Department of Entomology and Plant Pathology, North Carolina State University , Raleigh , NC , United States of America
6 College of Science and Engineering, James Cook University , Townsville , Queensland , Australia
Andrew Nigel
Electronic publication date: 2024 Sep 24
Publication date: 2024
Volume: 12
Electronic Location ID: e17904
Received 2024 May 17; Accepted 2024 Aug 1
Copyright: ©2024 Justine et al.
Copyright year: 2024
Copyright holder: Justine et al.
License: This is an open access article distributed under the terms of the Creative Commons Attribution License, which permits unrestricted use, distribution, reproduction and adaptation in any medium and for any purpose provided that it is properly attributed. For attribution, the original author(s), title, publication source (PeerJ) and either DOI or URL of the article must be cited.
License URL: https://creativecommons.org/licenses/by/4.0/

Keywords: Platyhelminthes, Invasive alien species, USA, Mitogenome, Land flatworms, Taxonomy, New species

Funding: ISYEB, Institut de Systématique, Évolution, Biodiversité, Muséum National d’Histoire Naturelle, Paris, France This work was supported by ISYEB, Institut de Systématique, Évolution, Biodiversité, Muséum National d’Histoire Naturelle, Paris, France. The funders had no role in study design, data collection and analysis, decision to publish, or preparation of the manuscript.

==============================
Specimens of a flat and dark brown land planarian were found in a plant nursery in North Carolina, USA in 2020. On the basis of examination of photographs of the live specimens only, the specimens were considered as belonging to Obama nungara, a species originally from South America, which has now invaded a large part of Europe. Unexpectedly, a molecular analysis revealed that the specimens did not belong to this species, neither to the genus Obama. We then undertook its histological study, which finally confirmed that the species is a member of the genus Amaga: the species is herein described as a new species, Amaga pseudobama n. sp. The species has been found in three locations in North Carolina and some infested plants were from Georgia. We reinvestigated specimens collected in Florida in 2015 and found that they also belong to this species. Citizen science observations suggest its presence in other states. Therefore, it is likely that A. pseudobama has already invaded a part of south-east USA and that the invasion took place more than ten years ago. The complete 14,909 bp long mitochondrial genome was obtained. The mitogenome is colinear with those of other Geoplanidae and it was possible to find and annotate a tRNA-Thr, which has been reported missing in several geoplanids. Amaga pseudobama shares with other Geoplaninae the presence of alternative start codons in three protein-coding genes of its mitogenome. The availability of this new genome helped us to improve our annotations of the ND3 gene, for which an ATT start codon is now suggested. Also, the sequence of the ATP6 gene raised questions concerning the use of genetic code 9 to translate the protein-coding genes of Geoplanidae, as the whole translated protein would not contain a single methionine residue when using this code. Two maximum likelihood phylogenies were obtained from genomic data. The first one was based on concatenated alignments of the partial 28S, Elongation Factor 1-alpha (EF1) and cox1 genes. The second was obtained from a concatenated alignment of the mitochondrial proteins. Both strictly discriminate A. pseudobama from O. nungara and instead associate it with Amaga expatria. We note that the nine species currently accepted within Amaga can be separated into two groups, one with extrabulbar prostatic apparatus, including the type species A. amagensis, and one with intrabulbar prostatic apparatus, including the new species A. pseudobama. This suggests that species of the latter group should be separated from Amaga and constitute a new genus. This finding again illustrates the possible emergence of new invasive species in regions naturally devoid of large land planarians, such as North America. Amaga pseudobama thus deserves to be monitored in the USA, although its superficial resemblance to O. nungara and Geoplana arkalabamensis will complicate the use of photographs obtained from citizen science. Our molecular information provides tools for this monitoring.

Introduction

In recent decades, many species of Geoplanidae have been reported as invasive species in most parts of the world (Justine et al., 2018; Justine et al., 2020b; Justine, Gastineau & Winsor, in press; Sluys, 2016). Some of these species have very characteristic coloured patterns that allow certain identification, even from sometimes imperfect photos collected using citizen science. Others have coloured patterns that are much more difficult to identify with certainty. This is particularly the case of Obama nungara (Carbayo et al., 2016), whose general body colour mostly varies between dark brown and almost black, and whose general body shape (flat, with a tapered anterior end) corresponds to that of many species. As a result, the identification of the species Obama nungara was controversial in the early 2010s, due to confusion with Obama marmorata which resembles it (Lago-Barcia et al., 2015). Only a detailed anatomical study and molecular data made it possible to definitively separate the two species (Carbayo et al., 2016). Obama nungara, native to Brazil and Argentina, is now reported in many countries in Europe (Čapka & Čejka, 2021; Justine et al., 2020b; Justine, Gastineau & Winsor, in press; Lago-Barcia et al., 2019; Mori et al., 2022; Soors et al., 2019; Thunnissen et al., 2022), islands such as the Azores in the Atlantic (Lago-Barcia, González-López & Fernández-Álvarez, 2020) and La Réunion in the Indian Ocean (Justine et al., 2022b), and a few records can be found in iNaturalist for North America. Ecological niche modelling has shown that the species has the potential to invade a significant portion of the world (Fourcade, 2021; Negrete et al., 2020).

Specimens of land planarians were found in a plant nursery in North Carolina, USA in July 2020. On the basis of examination of photographs of the live specimens only, the specimens were identified as Obama nungara. To our surprise, a molecular analysis showed that the species was very different. We then carried out a detailed morphological and histological analysis, reported here, which allows us to describe these specimens as a new species, Amaga pseudobama n. sp. We also report new information based on sequencing various genes of the new species, including its complete mitogenome.

This misidentification highlights the need to avoid basing biogeographic analyses solely on photos of a putative species, as is often the case in citizen science work, but also to examine representative specimens when there is a risk of confusing one species with another of similar appearance.

Materials and Methods

Specimens

Specimens from North Carolina, USA

In July 2020, a plant nursery in Kinston, Lenoir County, North Carolina, purchased a lot of 600 pots of ornamental plants from a different nursery in Georgia, USA. Flatworms were discovered on pots during repotting. Specimens were examined by one of us (MAB), who took photographs of livings specimens, then transferred three of them to DGR at the Academy of Natural Sciences of Drexel University. Photographs of specimens were sent to one of us (LW) who identified the species as Obama nungara. It was decided to examine the specimens and to proceed to molecular analysis to confirm this provisional identification. Three specimens were sent to one of us (JLJ) for identification. Specimens were registered in the collections of the Muséum national d’Histoire naturelle, Paris, as MNHN JL380A, B, and C. Due to the COVID-19 pandemic and shutdowns of laboratories, routine molecular analysis were delayed. In 2022, half of one specimen was sent to RG for a molecular study; when results showed that the species was not O. nungara, one specimen was transferred to LW for morphological and histological examination.

In addition to Kinston, similar specimens were also found in Charlotte, Mecklenburg County, North Carolina, in September 2020, and in Locus, Stanley County, North Carolina, in July 2021; in both cases, MAB took photographs of living specimens.

Specimens from Florida, USA

Specimens from Florida were collected by the late John Herman from the Florida Gulf Coast University and his students in 2015 and forwarded to the MNHN in Paris, where they were registered and barcoded. Other specimens collected by this team included Platydemus manokwari (Justine et al., 2021).

Specimens were (listed as MNHN registration number, date, time, locality, collector, GenBank registration number for partial cox1): MNHN JL276A & MNHN JL276B (2 specimens), 23 September 2015, 10:30, North Fort Myers, coll. Tesla Richards, GenBank: PP766724 and PP766725; MNHN JL277 (1 specimen), 23 September 2015, 20:00, Cobblestone on the Lake, Fort Myers, coll. Damian Baker, GenBank: PP766726; MNHN JL278 (1 specimen), 24 September 2015, 8:24, Cobblestone on the Lake, Fort Myers, coll. Damian Baker, GenBank: PP766727.

All four specimens were barcoded for the cox1 gene with Sanger sequencing, according to routine methods (Justine et al., 2015).

Histology

The specimen JB380B was divided into anterior, pre-pharyngeal, and posterior pieces, dehydrated in an ascending ethanol series to 90% ethanol, cleared in terpineol (the eyes were examined in the cleared specimen at this point), then infiltrated and the three pieces embedded as a single block, in Leica Biosystems Surgipath Paraplast paraffin–plastic polymer wax, melting point 56 °C (Winsor & Sluys, 2018). The block was serially sectioned at 7 µm thickness from the left side on a Leitz 1212 rotary microtome. Sections were mounted on microslides treated with Mayer’s egg albumen section adhesive (Winsor & Sluys, 2018), stained together with a staining fidelity control slide (Caenoplana variegata TS) with Steedman’s Triple stain (Steedman, 1970), and Mayer’s Haemalum and Eosin Y (Kiernan, 2008), then mounted in Entellan New mounting medium (Merck). Descriptors for colour matches are taken from the RAL colour chart at https://www.ralcolorchart.com. Calculation of the Subepidermal Musculature Index (SMI—subepidermal musculature thickness: body height ratio) was calculated at the pre-pharyngeal region according to Froehlich (1954). The Copulatory Apparatus Index (CAI): the ratio of length of copulatory apparatus to body length measured from proximal portion of prostatic vesicle towards distalmost part of the female atrium, follows Negrete et al. (2022).

Next generation sequencing

A part of specimen MNHN JL380A was sent to the Beijing Genomics Institute in a sealed tube with 90% ethanol (BGI-Shenzhen, Shenzhen, China). The sequencing process was conducted by BGI and the library produced was sequenced on a DNBSEQ platform. A total of ca. 34 million clean 150 bp paired-end reads were obtained, and assembled using SPAdes 3.15.5 (Bankevich et al., 2012) with a k-mer parameter of 125. The mitochondrial genome and rRNA genes were datamined from the contigs file by standalone BLAST analyses (Camacho et al., 2009) using reference sequences from Amaga expatria as database. Annotations were done with the help of MITOS (Bernt et al., 2013) using genetic code 9, except for the mitochondrial rRNA genes which required alignments with the corresponding genes of A. expatria. The position of the tRNA was also verified using ARWEN v1.2 (Laslett & Canbäck, 2008). The map of the mitogenome was drawn with OGDRAW (Lohse et al., 2013). The EF1-α gene was retrieved by datamining the contigs files obtained on both species of Amaga spp. by standalone blastx analyses (Camacho et al., 2009) with the protein sequence of Choeradoplana pucupucu Carbayo, Silva, Riutort & Alvarez-Presas, 2017 (AWI47783) as database and a e-value filter of 1e−100. The 28S gene was also retrieved using standalone BLAST analyses, with the partial 28S gene of A. expatria (MZ562764) as reference.

Molecular phylogeny

The three-gene phylogeny was produced with respect to the recent comprehensive phylogeny of Geoplaninae provided in the article describing Timyma olmuensis Almeida & Carbayo, 2021 (Almeida et al., 2021). This maximum likelihood phylogeny was based on concatenated alignments of three genes, namely the mitochondrial gene of the cytochrome c oxidase subunit 1 (cox1 or COI) the nuclear ribosomal RNA gene for the large subunit (28S or LSU) and the nuclear gene of the Elongation Factor 1-alpha (EF1-α).

The mitochondrial protein phylogeny was produced from a matrix made by appending a recently published multiprotein dataset (Gastineau et al., 2024) with data from the new species.

Genes and proteins were aligned separately using MAFFT 7 (Katoh & Standley, 2013) and trimmed with trimAl (Capella-Gutiérrez, Silla-Martínez & Gabaldón, 2009) with the -automated1 option. The best model of evolution was evaluated on each distinct alignment with ModelTest-NG v0.1.7 (Darriba et al., 2019) with default option, or with the whole concatenated alignment in the case of the multiprotein phylogeny. Alignments were concatenated using Phyutility 2.7.1 (Smith & Dunn, 2008). The maximum likelihood (ML) phylogenetic analysis was conducted using IQ-TREE 2.2.0 (Minh et al., 2020) and 1,000 ultrafast bootstrap replicates, with a dataset partitioned by best models of evolution which were TPM2uf+I+G4 for 28S, GTR+I+G4 for cox1 and GTR+G4 for EF1. For the multiprotein phylogeny, the best model of evolution was MTZOA+I+G4+F and 1,000 ultrafast bootstrap replicates were computed. The outgroup for the three-genes tree consisted in Dugesia sicula Lepori, 1948, Dugesia deharvengi Kawakatsu & Mitchell, 1989 and Dugesia subtentaculata (Draparnaud, 1801), while the outgroup for the multiprotein tree was Prosthiostomum siphunculus (Delle Chiaje, 1822) (KT363736).

ZooBank registration

The electronic version of this article in Portable Document Format (PDF) will represent a published work according to the International Commission on Zoological Nomenclature (ICZN), and hence the new names contained in the electronic version are effectively published under that Code from the electronic edition alone. This published work and the nomenclatural acts it contains have been registered in ZooBank, the online registration system for the ICZN. The ZooBank LSIDs (Life Science Identifiers) can be resolved and the associated information viewed through any standard web browser by appending the LSID to the prefix http://zoobank.org/. The LSID for this publication is: urn:lsid:zoobank.org:pub:49F66ABC-EEAA-4C92-8CCF-6B9200AB91B6. The online version of this work is archived and available from the following digital repositories: PeerJ, PubMed Central SCIE and CLOCKSS.

Results

Morphological results and description of the new species

Systematics:	
Order Tricladida Lang, 1884	
Family Geoplanidae Stimpson, 1858	
Subfamily Geoplaninae Stimpson, 1858	
Genus AmagaOgren & Kawakatsu, 1990	

Amaga pseudobama n. sp.

Etymology

The specific epithet alludes to the species being initially identified from photographs as the darkly pigmented form of Obama nungara.

Material examined

Material consists of three specimens from North Carolina, USA (type-material) and 4 specimens from Florida, USA (vouchers).

Type-material: Three specimens, all collected in Kinston, Lenoir County, North Carolina, USA on 27 July 2020 and fixed in ethanol. Holotype, MNHN JL380B, small incision on side. Longitudinal sagittal sections of the anterior end, transverse sections of the pre-pharyngeal portion, and longitudinal sagittal sections of the posterior end incorporating the pharynx and copulatory organs, slides 1–75 stained with Steedman’s Triple stain, and slides 76–80 stained with Mayer’s Haemalum and Eosin Y. Paratype, MNHN JL380A, hologenophore, not sectioned, half of body destroyed for molecular analysis. Paratype, MNHN JL380C, not sectioned, body coiled.

Vouchers: Four specimens collected in two localities in Fort Myers, Florida, in September 2015, MNHN JL276A, MNHN JL276B, MNHN JL277, MNHN JL278. All specimens with small incision on side, all individually barcoded (cox1), sequences deposited as GenBank PP766724 –PP766727.

Type-locality: Kinston, Lenoir County, North Carolina, USA.

Living specimens

A living specimen from Kinston, North Carolina (Fig. 1), is broadly lanceolate with tapered extremities, and shallow convex in cross section. The unevenly coarsely mottled dorsal ground colour is black-brown (RAL8022) but looks almost black, grading laterally to terra-brown (RAL8028). A thin whitish mid-dorsal longitudinal stripe is present (visible in the anterior third of Fig. 1B). The ventral ground colour is finely mottled terra-brown, separated by a pale whitish median stripe that extends from the anterior end to the mouth where it broadens over the pharyngeal area and gonopore, after which it thins and disappears into the ground colour.

Figure 1 Amaga pseudobama n. sp., living specimen collected in Kingston, North Carolina in July 2020.

The specimen was photographed in a Petri dish on a white background. (A) Dorsal view from back; the head is at top. (B) Dorsal view; the head is on the right. (C) Ventral view; the pharynx (p) and gonopore (g), are indicated; the head is on the left. Unscaled. Photographs by Matthew A. Bertone.

Diagnosis

Body broadly lanceolate in shape with rounded extremities (preserved specimen). Dorsal ground colour unevenly coarsely mottled terra-brown anteriorly grading to black-brown posteriorly, with a whitish mid-dorsal longitudinal stripe and white body margins, and with two broad reddish-brown ventral longitudinal stripes (these are clearly evident on the fixed specimen but are not as obvious on the living specimen). With numerous small monolobulated ocelli and less numerous large bilobulated ocelli; eyes contour the anterior tip in a single row, crowd dorsolaterally, extending some two-thirds along the body, sparsely in the posterior third. Dorsally, eyes exhibit haloing (a clear space surrounding the eye), and dark body pigment around the underlying testes; SMI 4.6–5.6; pharynx cylindrical type; oesophagus present; oesophagus-pharynx ratio 1:1.8; spermiducal vesicles join to form a common sperm duct that enters a tubular intrabulbar S-shaped seminal vesicle, that opens into the prostatic vesicle; eversible penis; male atrium highly folded ventrally; ovovitelline ducts with postflex condition with dorsal approach, receives shell gland secretions in terminal portion, separately enter female glandular canal. Female atrium capacious.

External morphology and colour pattern

The holotype measured 19.0 mm long, 3.6 mm wide at the mouth, and 1.2 mm thick, with the mouth 12.0 mm, and gonopore 14.4 mm, from the anterior end (Table 1).

Table 1 Morphological parameters of preserved specimens of Amaga pseudobama n. sp.

Parameter and units	Holotype
JL380B	Voucher
JL276A	Voucher
JL277	Voucher
JL278	
Molecular information, cox1 in GenBank		PP766724	PP766726	PP766727	
Locality	Kinston,
North Carolina	Fort Myers,
Florida	Fort Myers,
Florida	Fort Myers,
Florida	
Length (mm)	19.0	27.8	21.6	22.5	
Width at the mouth (mm)	2.5	5.0	4.3	4.4	
Height (μm)	1,030	–	–	–	
Mouth (mm) from the anterior tip, and as a % of the total length	11.5
60.5%	15.8
56.8%	11.7
54.2%	14.7
65.3%	
Gonopore (mm) from the anterior tip, and as a % of the total length	14.0
73.7%	20.1
72.3%	15.4
71.3%	17.7
78.7%	
Distance mouth - gonopore (mm), and as a % of the total length	2.5
13.2%	4.4
15.8%	3.7
17.1%	3.0
13.3%	
Creeping sole % body width	62%	–	–	–	
Subepithelial muscular index	4.6–5.6	–	–	–	

The preserved specimen is broadly lanceolate in shape with rounded extremities, and shallow convex in cross section. The dorsal ground colour is unevenly coarsely mottled, mainly terra-brown (RAL8028) in the anteriad quarter, grading to black-brown (RAL8022) in the posterior half. A whitish mid-dorsal longitudinal stripe with ill-defined margins, sparsely mottled with irregular brown pigment, occupies 15–23% of the body width for just over half the length of the dorsal surface, with residual pale median patches over the pharyngeal region and copulatory organs (Fig. 2A). The body margin is white, extending ventrally as a rim around the whole body (Fig. 2B). Two broad reddish-brown ventral longitudinal stripes with fuzzy lateral margins, each about a fifth to almost the full body width, and separated by a thin pale mid-ventral line, extend almost the entire length of the body (Fig. 2C). The eyes contour the anterior tip in a single row (Fig. 3A), and crowd dorsolaterally (Fig. 3B) extending some two-thirds along the body, but sparsely in the posterior third. Dorsally, the eyes are haloed by the dark body pigment (Figs. 3B, 3C).

Figure 2 Amaga pseudobama n. sp., general view of preserved holotype.

(A) Dorsal surface, (B) right side, and (C) ventral surface showing mouth (m) and gonopore (gp).

Figure 3 Amaga pseudobama n. sp., eyes of holotype.

(A) Eyes (arrowed) contouring the anterior tip, (B) eyes extending dorso-laterally, many of which are haloed (arrowed; a clear space surrounding the eye), (C) dorsum showing haloed eyes, and pigment surrounding the underlying testes (arrowed). Unscaled.

Body wall and musculature (Fig. 4)

The epithelium is thinner dorsally (16–18 µm) than ventrally (37.8–46.8 µm). Two types of rhabdoids, discharged from mesenchymal rhabditogen cells, are present: erythrophil rhammites measuring 12.6 µm × 1.4 µm–28.7 µm × 2.8 µm (length × width) present dorsally, and extending to the outer ventral zone where they are most numerous. Numerous erythrophil micro rhabdites measuring 4.6 µm × 1.4 µm–8.4 µm × 2.1 µm (length × width) are present in the ciliated ventral epithelium. Granular erythrophil and amorphous cyanophil secretions, derived from mesenchymal glands, are sparse in the dorsal epithelium, but are numerous in the marginal zone. Copious amorphous secretions pass from mesenchymal cyanophil glands through the ventral epithelium, together with relatively sparse granular erythrophil secretions.

Figure 4 Amaga pseudobama n. sp., prepharyngeal region, histology.

Transverse section of the pre-pharyngeal region. In order to ensure adequate resolution of the histology, just over a half of the section is shown.

Dark brown granular pigment is scattered throughout the dorsal mesenchyme but is concentrated around the testicular margins and lightly over the dorsal side of the testes.

The well-developed cutaneous musculature is tripartite, thicker ventrally than dorsally, comprising a single fibre thickness of circular muscle, then paired decussate fibres, and longitudinal muscles in bundles of 4–5 fibres dorsally, 6–8 fibres ventrally, SMI 4.6–5.6. The muscle fibres in cross section are solid, not hollow. The subepidermal musculature is not insunk anteriorly or elsewhere.

The parenchymal musculature is weak, and comprises a few fibres of dorsal transverse muscles, sparse supraintestinal muscles, and more numerous infraintestinal and supraneural transverse muscles. The oblique dorsoventral muscles, and peri-intestinal muscles are prominent. Parenchymal longitudinal muscles are absent. There are no anterior muscular specialisations.

Alimentary system

The pharynx is approximately 840 µm long, cylindrical with a ruffled tip (Fig. 5), with the dorsal insertion before the mouth, and posterior to the ventral insertion. The lip of the pharynx is partly folded back into the lumen. The inner pharynx is lined by a nucleate, ciliated secretory epithelium. The underlying inner pharyngeal musculature comprises a few fibres of longitudinal muscle, external to which are circular muscles interspersed with a few longitudinal muscles (mixed musculature). The parenchymatous mid-pharyngeal region is comprised of sparse radial and longitudinal muscle fibres, together with secretory ducts containing erythrophil and cyanophil granules. The former secretions are discharged at the tip of the pharynx, and the latter into the lumen. The outer pharyngeal wall comprises an outer low ciliated cuboidal epithelium underlain by a few longitudinal muscle fibres, internal to which are weak circular muscles bounded by the sparse longitudinal fibres of the mid-pharyngeal wall.

Figure 5 Amaga pseudobama n. sp., pharynx, histology.

(A) Pharynx, longitudinal sagittal section, and (B) more laterally, showing the folded pharyngeal lip.

The pharyngeal pouch is 1.4 mm long, representing 7.4% of the body length. The mouth is situated in the ventro-posterior third of the pouch. An oesophagus is present, measuring some 460 µm with the ratio of the length of the oesophagus to pharynx length 1:1.8.

The gastrodermis is vacuolate, and the intestinal lumen is mostly empty. Gregarines were not observed in the gut or elsewhere.

Sensory organs

The unpigmented sensory zone contours the sub-margin of the anterior tip and contains simple ciliated pits 10.8 µm–12.6 µm deep and 12.6 µm–14.4 µm wide, with an interval of 28 µm–34 µm between them (Fig. 6).

Figure 6 Amaga pseudobama n. sp., anterior region, histology.

Anterior tip, longitudinal lateral section, showing mono- and bilobulated eyes, and ciliated pits.

There are two types of eyes: numerous small monolobate ocelli 14.4 µm × 19.8 µm–28 µm diameter, and less numerous large bilobate ocelli 36–40 µm × 21.6–25.2 µm (Fig. 6).

Reproductive organs

The copulatory organs (Figs. 7 and 8) lie 1.3 mm behind the pharyngeal pouch, measuring 1,780 µm long, and 900 µm high, with a length-height ratio of almost 2:1. The Copulatory Apparatus Index (CAI) body length-length of copulatory apparatus is 10.8:1. The lengths of the male and female atria, based upon their cytology, are almost equal. The penis is the eversible type. The gonopore is patent and is situated in the posteriad third of the copulatory apparatus. The penis bulb consists of a thin outer muscularis surrounding an open meshwork of weak longitudinal fibres that would allow compression of the bulb when the strongly muscularised male atrium contracts to evert the penis during copulation.

Figure 7 Amaga pseudobama n. sp., reconstruction of reproductive organs.

Copulatory organs, composite reconstruction, longitudinal sagittal aspect. Some details of secretions and musculature have been omitted for clarity. The dashed line is the approximate area of transition between the male and female atria.

Figure 8 Amaga pseudobama n. sp., copulatory organs, histology.

Copulatory organs, longitudinal sagittal section. Secretion and musculature omitted from the composite reconstruction can be clearly seen.

The ellipsoid testes, measuring 135 µm–205 µm in antero-posterior diameter, contain mature sperm, and are situated just above the very weak supra-intestinal transverse muscle layer, extending in irregular staggered rows 2.3 mm from the anterior tip to the root of the pharynx. Dorsolaterally they are delineated by a thin layer of dark pigment (Fig. 9). A sperm duct (vas efferens) passes ventrally from the distal pole of each testis to the vasa deferentia situated dorsad to the ovovitelline ducts lying just above the ventral nerve cords. The vasa deferentia, with a lumen of approximately 34 µm–35 µm in diameter lined by a flattened nucleate epithelium, bounded by a very thin muscularis, pass posteriorly. Some 440 µm behind the pharyngeal pouch, the vasa deferentia greatly distend forming spermiducal vesicles, with an internal diameter of 43 µm–44 µm bounded by a very thin muscularis and filled with mature sperm. Just anteriad to the penis bulb, the spermiducal vesicles ascend dorsomedially through the outermost longitudinal-oblique bulb musculature and unite. Up to this point, there is no evidence that the spermiducal vesicles receive secretions from glands in the surrounding parenchyma.

Figure 9 Amaga pseudobama n. sp., testis, histology.

Testis, located above the supraintestinal transverse parenchymal muscles. Note the dark pigment that delineates the testes.

The short common spermiducal vesicle opens into the proximal end of a tubular S-shaped seminal duct, internal diameter of 60 µm, bounded by a few thick oblique muscle fibres, and situated in outer muscularis of the penis bulb. The duct is lined by a nucleate ciliated cuboidal epithelium that receives predominantly fine cyanophil granular secretions, and relatively sparse fine–medium granular pale erythrophil secretions from glands in the surrounding parenchyma.

The seminal duct rises dorsomedially within the open reticulate mesh of weak longitudinal-oblique fibres forming the muscularis of the penis bulb, to enter the antero-dorsal aspect of the prostatic vesicle towards its left side (Fig. 10A). There is no evidence at this point of a papilla or papilla-like fold in the epithelium; nor is there any specialised organisation of musculature underlying the epithelium suggesting a papilla.

Figure 10 Amaga pseudobama n. sp., male organs, histology.

(A) The common sperm duct and seminal vesicle that arise laterally to the prostatic vesicle. The dashed line indicates the approximate junction of these vesicles; (B) details of the start of the opening of the slit in the prostatic vesicle, together with detail of the erythrophil secretions, and the cyanophil secretions of the male atrium.

The epithelium of the prostatic vesicle is densely charged with coarse strongly erythrophil granules together with numerous granular cyanophil secretions; both discharge in a merocrine manner through a ciliated columnar epithelium into the lumen via elongated gland necks. Both secretions are derived from glands in the surrounding parenchyma. The prostatic vesicle empties into the male atrium via a progressively enlarged vertical slit in the sulcus, starting from the upper left side (Fig. 10B). There is no obvious papilla at a point where there is a sharp transition between the epithelium of the prostatic vesicle and that of the atrium. The weak musculature of the prostatic vesicle is similar to that of the seminal vesicle.

The atrial musculature comprises a single inner layer of strong circular, then outer strong longitudinal muscles underlying the lining epithelium. The ventral surface and walls of the male atrium are folded to create three main lateral diverticula or sulci when the penis is not everted, and into which the atrial secretions are discharged. The dorsal atrial wall is largely unfolded. The two anteriad diverticula receive predominantly strongly cyanophil coarsely granular secretions, and finer lightly cyanophil granular secretions through a nucleate columnar epithelium. The third diverticulum merges with the female atrium (approximate transition marked by a dashed line in Fig. 11) and is lined by a tall nucleate columnar epithelium through which predominantly fine cyanophil granules are discharged that condense to form cyanophil strands in the lumen. The dorsal atrial wall is lined by a columnar epithelium with secretions approximating those of the ventral atrial epithelium opposite. The female atrium is characterised by an epithelium with the necessary precursor secretions to form an egg-capsule (de Souza & Leal-Zanchet, 2004; Shinn, 1993; Winsor, 1998): a tall nucleate columnar epithelium through which is discharged membrane bound chromophobe globules, fusiform erythrophil membrane bound packets, and amorphous lightly cyanophil secretions that form strands to which the chromophobe globules and erythrophil attach.

Figure 11 Amaga pseudobama n. sp., female organs, histology.

Detail of the female atrium characterised by an epithelium with the necessary precursor secretions to form an egg-capsule: a tall nucleate columnar epithelium through which is discharged membrane bound chromophobe globules, fusiform erythrophil membrane bound packets, and amorphous lightly cyanophil secretions that form strands to which the chromophobe globules and erythrophil attach (arrowed). The dashed line is the approximate area of transition between the male and female atria.

The ovaries are located 3.5 mm (right) and 4.0 mm (left) from the anterior tip and are nestled in the dorsal aspect of the ventral nerve cords. They are both of similar fusiform shape and size, measuring 570 µm long, 150 µm maximum height, and 200 µm maximum width (Fig. 12A). The ovovitelline ducts, lined by a nucleate, ciliated columnar epithelium emerge from the mid-dorsal wall of each ovary and pass posteriorly just above the ventral nerve cords. Vitellaria occupy the space surrounding the gut diverticula and discharge their secretions via short ciliated vitelline funnels into the ovovitelline duct. Just before the gonopore, the ovovitelline ducts curve medially and ascend at an angle of approximately 45° to separately enter the proximal female glandular canal: the postflex condition with dorsal approach. Deeply erythrophil shell glands discharge into the terminal portions of the ovovitelline ducts and into the proximal end of the female glandular canal (Fig. 12B). The mid-section of the glandular canal is lined by a tall ciliated nucleate columnar epithelium through which pass membrane bound cylindroid packets of amorphous strongly erythrophil secretions from the surrounding mesenchyme. In the distal section of the canal, the epithelium transitions into that of the female atrium. The glandular canal is bounded by the inner circular and outer longitudinal muscles surrounding the female atrium. Adenodactyls, adenomuralia, resorptive vesicles, and viscid gland are absent.

Figure 12 Amaga pseudobama n. sp., female organs, histology.

(A) Right ovary and ovovitelline duct; (B) separate entry of the ovovitelline ducts from either side of the proximal end of the female glandular canal.

Other similar specimens in the USA

Based on collected specimens only, we have proof that the species was present in 2015 in Florida and in 2020 in North Carolina.

The plant nursery in Kinston, North Carolina, was asked in 2022 if any other specimens had been found and answered negatively. Since the potted plants originally came from another nursery in Georgia, we can speculate that at least these two states are invaded by A. pseudobama. Flat, brown specimens of land flatworms have been reported in various states of the USA on iNaturalist; however, their identity cannot be ascertained, since these specimens could be A. pseudobama, but could also be small specimens of Obama nungara or specimens of Geoplana arkalabamensis.

We present in Table 2 a list of observations that were originally identified as either Obama nungara or Geoplana arkalabamensis in iNaturalist (January 2024). We examined the photographs, based on the additional information we now have, specifically size, eyes (presence or absence), and haloing of the eyes. We found that some identifications were probably erroneous. We consider that two observations in Texas and California, respectively, could be attributed to A. pseudobama. See the discussion about the difficulties in identifying these species from photographs of dorsal views only, generally without a precise scale.

Table 2 Records of geoplanid specimens originally identified as either Obama nungara or Geoplana arkalabamensis in iNaturalist (January 2024) in Southern USA and Mexico.

We re-examined the photographs and show that some specimens were probably misidentified, emphasizing the difficulties in correctly differentiating the three species O. nungara, G. arkalabamensis and A. pseudobama from photographs. Additionally, if our new identifications are correct, A. pseudobama could be present in California and Texas in addition to our records from specimens in North Carolina and Georgia.

iNaturalist Record	State	Date	Size	Original id in iNaturalist	Revised id	
https://www.inaturalist.org/observations/141651161	CA	Nov 10, 2022	30 mm	O. nungara ?	G. arkalabamensis	
https://www.inaturalist.org/observations/117066453	TX	March 10, 2022		O. nungara	G. arkalabamensis	
https://www.inaturalist.org/observations/101784748	CA	Nov 22, 2021	10–25 mm	O. nungara	Amaga pseudobama	
https://www.inaturalist.org/observations/101583366	TX	Nov 16, 2021	Ca. 2 inches	O. nungara	Probably G. arkalabamensis	
https://www.inaturalist.org/observations/101380433	LA	Nov 15, 2021	?	O. nungara	Possibly G. arkalabamensis	
https://www.inaturalist.org/observations/97329911	TX	October 2021		O. nungara	Amaga pseudobama	
https://www.inaturalist.org/observations/96429853	Mexico	Sep 27, 2021	6 cm	O. nungara	Geoplanidae	
https://www.inaturalist.org/observations/91524357	LA	Aug 17, 2021 	?	O. nungara	Geoplanidae	
https://www.inaturalist.org/observations/81786454	SC *	Jun 4, 2021	Small	O. nungara	G. arkalabamensis	
https://www.inaturalist.org/observations/75611915	SC *	Apr 28, 2021	Small	O. nungara	G. arkalabamensis	
https://www.inaturalist.org/observations/72134429	SC *	Mar 26, 2021	Small	O. nungara	G. arkalabamensis	
https://www.inaturalist.org/observations/68472771	NC	Nov 23, 2020		O. nungara	O. nungara	
https://www.inaturalist.org/observations/146667528	FL	Jan 15, 2023	1,5 inch or 1, 5 cm ?	G. arkalabamensis	G. arkalabamensis	
https://www.inaturalist.org/observations/144074412	FL	Dec 10, 2022	?	G. arkalabamensis	Platydemus manokwari	
https://www.inaturalist.org/observations/20363442	AL	Feb 15, 2019	?	G. arkalabamensis	Probably G. arkalabamensis	
https://www.inaturalist.org/observations/15281507	AL	Aug 9, 2018	?	G. arkalabamensis	G. arkalabamensis	
https://www.inaturalist.org/observations/8555997	AL	Aug 25, 2008	4 cm	G. arkalabamensis	G. arkalabamensis	
https://www.inaturalist.org/observations/8489856	AL	Oct 21, 2017	small	G. arkalabamensis **	G. arkalabamensis	
https://www.inaturalist.org/observations/1382046	AL	Apr 10, 2015	?	G. arkalabamensis	G. arkalabamensis	

Molecular results

Specimens from Florida vs North Carolina

The cox1 partial sequences of all 4 specimens from Florida (MNHN JL276A, JL276B, JL277, JL278) were identical with the sequence of the specimen from North Carolina (MNHN JL380A), thus ascertaining that all specimens belong to the same species.

The mitochondrial genome

A 14,909 bp contig was found after assembly. It displayed identical endings, but these endings were TATATAT motifs. Because of this, it is difficult to assess whether or not the genome was complete, or if the assembly stopped because of repeated sequences that cannot be resolved using short-reads sequencing (Gastineau et al., 2024). However, the contig contained all the conserved genes and for easier reading, it is depicted as circular in Fig. 13. The mitogenome (GenBank: PP727122) encodes for 12 protein coding genes, 2 rRNA and 22 tRNA and is colinear with those of other Geoplanidae. ND4L and ND4 overlap by 32 bp. It was possible to find and annotate a tRNA-Thr, which has been reported as missing among several species of Geoplanidae (Gastineau, Winsor & Justine, 2022; Justine et al., 2022a; Soo et al., 2023). However, there are two points that deserve to be underlined: (i) this tRNA is missing its D-Loop; and (ii) ARWEN completely failed to find it, while only MITOS suggested its presence.

Figure 13 Map of the mitochondrial genome of Amaga pseudobama n. sp.

The mitogenome is 14,909 bp long and contains 12 protein coding genes, two ribosomal RNA genes and 22 transfer RNA genes. Canonical start codons for the genetic code 9 could not be found for ATP6, cox2 and ND3.

As already observed with all Geoplaninae for which a mitogenome has been sequenced, it was impossible to find start codons for several protein coding genes using genetic code 9 and the alternative codons known to it. The genes concerned were ATP6, cox2 and ND3, a situation identical to that observed in A. expatria. As for A. expatria, the start codon for ATP6 and cox2 seems to be TTG. For ND3, we previously used TTA as a start codon when annotating the mitogenome of A. expatria. This choice was made after alignment with the reference sequence from O. nungara (KP208777) (Solà et al., 2015), in which the gene was annotated with a TTG as a start. As the number of available mitogenomes of Geoplaninae has increased, a different approach was tried to annotate this specific gene. Sequences from the three species of Geoplaninae were submitted to ORFfinder (Wheeler et al., 2003) with the ‘any sense codon’ option. This option is described as ‘find all stop-to-stop ORFs’. The corresponding nucleotide sequences retrieved for the ND3 gene were 357 bp, 411 bp and 408 bp for A. pseudobama, A. expatria and O. nungara, respectively. These three sequences aligned on 5′ starting with an ATT codon, which also represents the maximum extent of the sequence obtained from A. pseudobama. For these reasons, we chose ATT as the start codon for ND3.

Phylogenetic dataset and maximum likelihood phylogeny

It was easier to retrieve the Elongation Factor 1-alpha gene (EF1-α) for A. pseudobama (GenBank: PP722687) than for A. expatria (GenBank: PP729153). For A. pseudobama, we could obtain a complete 1,392 bp gene (including stop codon) while for A. expatria, we had to restrict ourselves to a 358 bp fragment. The sequence of the 28S gene corresponds to a 1,530 bp fragment that was assembled with a coverage of 730.45×. Reports suggest that this gene exists in two divergent types among Geoplanidae (Carranza et al., 1996; Gastineau et al., 2024), and it is difficult to assess the impact that mixing variants of this gene might have on a phylogeny. The concatenated alignment was 2,550 bp long, distributed 1,266 bp for 28S, 678 bp for cox1 and 612 bp for EF1-α, so the 28S gene accounted for half of it.

The three-gene phylogeny (Fig. 14) associated A. pseudobama and A. expatria in a cluster with maximum support at the node and discriminates it from the cluster containing Obama nungara (Carbayo et al., 2016) and Obama josefi (Carbayo & Leal-Zanchet, 2001).

Figure 14 Three-gene phylogenetic tree.

The tree is based on concatenated alignments of the 28S, cox1 and EF1α genes. Maximum likelihood phylogenetic tree based on 72 taxa, using the TPM2uf+I+G4, GTR+I+G4 and GTR+G4 models of evolution for 28S, cox1 and EF1, respectively. The tree was computed for 1,000 ultrafast bootstrap replicates; ML bootstrap support values indicated at the nodes. The closest species to Amaga pseudobama n. sp. is Amaga expatria.

The multiprotein phylogeny (Fig. 15) associated A. pseudobama with A. expatria and differentiated it from O. nungara, within a robust clade including these three Geoplaninae.

Figure 15 Maximum likelihood phylogenetic tree of concatenated mitochondrial proteins.

The tree is based on concatenated protein sequences extracted from 28 mitogenomes using the MTZOA+I+G4+F model and with 1000 ultrafast bootstrap replicates; ML bootstrap support values indicated at the nodes. The closest species to Amaga pseudobama n. sp. is Amaga expatria.

Discussion

Morphology and systematics

Citizen Science platforms, such as iNaturalist, are valuable resources, especially as they can facilitate the identification and distribution of invasive plant and animal species. However, identification of some species solely from photographs has limitations. For example, differentiating between species of exotic terrestrial flatworms with similar overall appearance. Ideally, identifications of such problematic species need to be confirmed by anatomical or molecular studies on representative specimens, as we did for the species described here.

The genus Amaga Ogren & Kawakatsu, 1990 presently comprises a heterogeneous group of eight species: Amaga amagensis (Fuhrmann, 1914) (type-species) (Fuhrmann, 1914); A. becki (Fuhrmann, 1914); A. bussoni (Froehlich, 1959); A. contamanensis (Hyman, 1955); A. expatria Jones & Sterrer, 2005; A. libbieae (Du Bois-Reymond Marcus, 1958); A. ortizi (Fuhrmann, 1914); and A. righii (Froehlich & Froehlich, 1972). Amaga pseudobama is presently the smallest species in the genus.

Following their study of the type species A. amagensis, Grau et al. (2012) re-evaluated the characters of the genus, and accordingly emended the genus diagnosis as follows: “Geoplaninae of large broad and flattened body with well-developed glandular body margins. SMI: 5–7%. Testes placed above supra-intestinal parenchymal transverse muscular layer; male atrium folded, with ejaculatory duct opening through a small papilla-like fold; penis eversible; prostatic vesicle extrabulbar, bifurcate and elongate. Ovovitelline ducts approaching copulatory apparatus from anterodorsal aspect and opening at the same point into the vagina. No cephalic specialisations; subepidermal musculature not insunk. Parenchymal longitudinal musculature absent. Adenodactyls or glandulo-muscular organs absent. Type species: Geoplana amagensis Fuhrmann, 1914”.

Externally, A. pseudobama is almost identical to Geoplana arkalabamensis Ogren & Darlington, 1991 (Geoplaninae), another invasive flatworm of unknown origin recorded in the USA (Ogren & Darlington, 1991). It has a dark brown dorsal ground colour with a pale indistinct median stripe, and exhibits haloing of the dorsal-lateral eyes. Mature specimens differ externally from A. pseudobama in exhibiting numerous small eyes in more than a single row contouring the anterior tip. However, juvenile specimens of G. arkalabamensis may exhibit a single row of eyes around the anterior tip, and combined with a small size could be confused with A. pseudobama. Many photographs of flatworm specimens on citizen science platforms are generally of insufficient focus and resolution to show anterior eye configuration. Internally, the two species differ significantly: G. arkalabamensis has a protrusible penis and a pronounced posterior diverticulum arising from the posteriad wall of the genital antrum, whereas A. pseudobama has an eversible penis, and no posterior diverticulum.

The presence of paired reddish brown ventral stripes visible in preserved specimens separate A. pseudobama externally from the other species presently assigned to Amaga, though the much larger species A. righii has a pale mid-dorsal stripe, and similar distribution of eyes. Similarities with A. righii are also noted in the general topography of the copulatory organs: vasa deferentia forming spermiducal vesicles that enter into what Froehlich & Froehlich (1972) term a “common male duct”, with thick muscularis, lined by a ciliated epithelium that receives fine-grained weakly eosinophil glands (in A. pseudobama, termed the seminal vesicle). This duct opens into the ental region of the male atrium forming “what could be called a prostatic apparatus”. Two glandular rings are present in the male atrium: pale eosinophil glands as in the common duct, and heavily stained eosinophil glands (in A. pseudobama, strongly erythrophil and strongly cyanophil secretions in adjacent sulci). The atrial musculature, especially the longitudinal muscles, is very strong over two-thirds of the proximal atrium, and the papilla-like fold in the male atrium is absent (also absent in A. pseudobama).

The species of Amaga can be divided into two groups on the basis of the presence and relative position of prostatic apparatus or prostatic vesicle in relation to the penis bulb. One group comprises those with an extra-bulbar prostatic vesicle, and the ejaculatory duct opening through a papilla-like fold that includes the type species Amaga amagensis, and A. becki, A. bussoni and A. ortizi. Prostatic apparatus is not identified in the description of A. libbieae though it does have a penial papilla.

The other group within Amaga consists of those taxa with an intrabulbar prostatic vesicle, and that lack an intra-penial papilla. The group includes Amaga contamanensis, A. expatria, A. righii and A. pseudobama. The presence of an intrabulbar prostatic vesicle, and absence of a papilla-like fold through which the ejaculatory duct opens, sets these four species of Amaga apart from the type species Amaga amagensis, and A. becki, A. bussoni and A. ortizi. However, the original description of A. expatria stated that the distal end of the penis was “highly convoluted and ill-defined” (Jones & Sterrer, 2005). It could be interpreted as a partly everted penis. A subsequent description based on a specimen from Martinique (Justine et al., 2020a), stated that the “ejaculatory duct terminated in a short penis”. Therefore, there is some uncertainty as to which of the two groups A. expatria belongs.

The species in this second group otherwise accord with the other characters that define Amaga: body shape, glandular margins, relative strength of the subepidermal musculature, testes positioned dorsal to the supra-intestinal transverse parenchymal muscles, folded male atrium, approach of the ovovitelline ducts to the copulatory apparatus and their entry into the female glandular duct, absence of cephalic specialisations, insunk subepidermal musculature, parenchymal longitudinal muscles, adenodactyls and glandular muscular organs.

Currently, with the exception of Amaga, it appears that no genus within the Geoplaninae includes species with extra-bulbar prostatic apparatus, together with species with intra-bulbar prostatic apparatus. There appears to be a reasonable case for possibly restricting Amaga to those species with extrabulbar prostatic apparatus (Amaga amagensis, and A. becki, A. bussoni and A. ortizi), and erecting a new genus to accommodate those Amaga species with intrabulbar prostatic apparatus. In the absence of molecular information on most species of Amaga, we currently refrain from doing so.

Of passing interest is the pigment surrounding the testes in A. pseudobama. This phenomenon has previously been observed in Amaga expatria (Jones & Sterrer, 2005), and in some other Geoplaninae including Obama ladislavii (von Graff, 1899) (E. M. Froehlich in Jones & Sterrer, 2005). Pigment surrounding testes is also found in Transandiplana graui Almeida, Álvarez-Presas & Carbayo, 2023 (Almeida, Álvarez-Presas & Carbayo, 2023).

Records from iNaturalist

Table 2 shows that very few Obama nungara-looking geoplanids are present in iNaturalist in North America. Before the publication of this paper and the description of A. pseudobama, specimens with a general brown dorsal colour and an Obama-looking shape could be identified either as G. arkalabamensis or O. nungara. Our revised identifications show that a single record could actually be O. nungara, and also suggest that A. pseudobama could be present in Texas and California, in addition to North Carolina, Georgia and Florida (based on our specimens). Obviously, Obama nungara-looking flatworm specimens found in the USA should be subjected to a molecular test on cox1 sequences to ascertain their identification.

Molecular results

Features of the mitochondrial genome

Amaga pseudobama is the second species of the genus Amaga for which a mitogenome has been sequenced, and only the third Geoplaninae after O. nungara (Solà et al., 2015). It is intriguing that the difficulties to find start codons concern all these species. The case of ATP6 is possibly the most exemplary. Using ORFfinder, the only way to find an ORF corresponding to this gene to appear was to choose the ‘any sense codon’ option, which is described as ‘find all stop-to-stop ORFs’. In the case of ATP6, this ‘maximum size’ ORF corresponds to the protein as annotated in A. pseudobama, meaning that there is also no possibility to extend the gene at its 5′ extremity to find a more suitable start codon. Moreover, and this constitutes a noticeable difference with A. expatria, the putative ATP synthase F0 subunit 6 protein of A. pseudobama contains in its whole sequence no other methionine residue that could have been considered as an alternative start for a shorter protein. With the exception of what has to be considered as the TTG-encoded initial methionine, the Atp6 protein of O. nungara possesses three methionine residues and in the case of A. expatria, this number is four. However, when trying to translate these genes with genetic code 5, in which TTG is a documented alternative start codon, the number of methionine residues (excluding again the putative first one) increases up to 11 for A. expatria and up to 18 for O. nungara and A. pseudobama. For now, we will continue to annotate the mitochondrial protein-coding genes of Geoplanidae using code 9, but as the number of available mitogenomes rises, we are more and more convinced that a verification/validation by protein sequencing would become necessary to remove the doubts.

Recent studies have unveiled unexpected conserved features in the mitogenomes of Geoplanidae, especially the extra-long cox2 gene found in the three species of Rhynchodeminae documented, namely Platydemus manokwari de Beauchamp, 1963 (Gastineau et al., 2020), Parakontikia ventrolineata (Dendy, 1892) Winsor 1991 (Gastineau & Justine, 2020) and Australopacifica atrata (Steel, 1897) (Gastineau, Winsor & Justine, 2022). The question of the validation and conservation of an alternative genetic code among the mitogenomes of Geoplaninae is in our opinion of equal importance if not higher, as the results show that at least two alternative start codons known in code 5 but not in code 9 have to be considered. The most recent classifications list up to 9 tribes and 37 genera in the Geoplaninae (Almeida et al., 2021; Carbayo et al., 2013). Therefore, our knowledge of complete mitogenomes is currently low in this subfamily, as it concerns only two genera. We strongly encourage additional sequencing of mitogenomes of Geoplaninae.

Molecular phylogeny

The three-gene tree and the multiprotein tree associate Amaga pseudobama with Amaga expatria and the clade formed by the two Amaga species is clearly differentiated, in both trees, from the branch of Obama nungara. Although these results need to be reinforced by sequences of additional geoplanins, they are sufficient to confirm that the new species belongs to Amaga, and not Obama.

Conclusion

It has been difficult to determine the affinities of this new species solely from morphological characters, as unfortunately, there is not always comparable treatment of comparable characters, nor uniformity in some of the terminology used in the descriptions of the species presently assigned to the genus Amaga. However, the molecular data support the inclusion of our species in Amaga.

Our study revealed the mitochondrial genome of a second species of Amaga, after Amaga expatria. This led to strongly reconsider some of the annotations formerly performed while confirming some others. Clearly, caution should be exercised when annotating the mitochondrial protein-coding genes of Geoplanidae, especially Geoplaninae. We hope that more data on additional species will clarify these issues in the future.

This work adds one species to the fauna of alien land flatworms present in the continental USA. This includes species whose presence has been known for a long time, such as the hammerhead flatworms Bipalium kewense Moseley, 1878, Bipalium adventitium Hyman, 1943 and Bipalium pennsylvanicum Ogren, 1987, and newcomers such as Platydemus manokwari (Hyman, 1943; Justine et al., 2015; Ogren, 1987). Most species can easily be distinguished from photographs. However, three species, Amaga pseudobama, Geoplana arkalabamensis and the dark morphs of Obama nungara, all three flat and dark species with bodies with similar general morphology, will be difficult to differentiate in citizen science photos. This poses a challenge and jeopardises the quality of the information that will be available on the invasion of the USA by these three species. However, the molecular information we provide here will help to differentiate the species, although currently no molecular information is available for G. arkalabamensis.

Disposable histological supplies were generously supplied by ProSciTech, Kirwan, Townsville, QLD, Australia to whom we are most grateful. We thank again the late John Herman and his students for their collections in Florida, USA.

Abbreviations in Figures

ce male atrial epithelium discharging cyanophil secretions

clm cutaneous longitudinal muscles

cm cutaneous musculature

cp ciliated pit

cs ciliated creeping sole

csd common spermiducal vesicle

cy cyanophil secretions

dfg distal female glandular canal (= vagina)

dip dorsal insertion of pharynx

dptm dorsal transverse parenchymal muscles

dtm dorsal transverse muscles

e eye/eyes

es erythrophil secretions

fa female atrium

fpl folded pharynx lip

g gonopore

gm glandular margin

i intestine

j junction of seminal and prostatic vesicles

le large bilobate eyes

m mouth

ma male atrium

nc nerve cord

ovd ovovitelline duct

pb penis bulb

pfg proximal female glandular canal

pfg proximal female glandular canal

ph pharynx

php pharyngeal pouch

pig pigment

pp penis papilla

pv prostatic vesicle

sd seminal duct

sdv spermiducal vesicle

se small monolobate eye

sg shell glands

slm subepithelial longitudinal muscle

te testis

tm transverse parenchymal muscle

vd vas deferens

vi vitellaria

vip ventral insertion of pharynx

Additional Information and Declarations

Competing Interests

Author Contributions

Data Availability

New Species Registration

Jean-Lou Justine is an Academic Editor for PeerJ.

Jean-Lou Justine conceived and designed the experiments, performed the experiments, analyzed the data, prepared figures and/or tables, authored or reviewed drafts of the article, obtained credits, coordinated the work, edited all figures, and approved the final draft.

Romain Gastineau conceived and designed the experiments, performed the experiments, analyzed the data, prepared figures and/or tables, authored or reviewed drafts of the article, performed genomics and phylogenetic analyses, and approved the final draft.

Delphine Gey performed the experiments, analyzed the data, authored or reviewed drafts of the article, performed Sanger sequencing, and approved the final draft.

David G. Robinson performed the experiments, analyzed the data, authored or reviewed drafts of the article, pre-identified specimens, and approved the final draft.

Matthew A. Bertone performed the experiments, analyzed the data, prepared figures and/or tables, authored or reviewed drafts of the article, collected specimens and made live photographs, and approved the final draft.

Leigh Winsor conceived and designed the experiments, performed the experiments, analyzed the data, prepared figures and/or tables, authored or reviewed drafts of the article, performed the histological study, and approved the final draft.

The following information was supplied regarding data availability:

The FASTA files and GBK file for the mitochondrial genome are available at Zenodo: Gastineau, R. (2024). A new species of alien land flatworm in the Southern United States [Data set]. Zenodo. https://doi.org/10.5281/zenodo.11164134.

The sequencing reads obtained from Amaga pseudobama are available on SRA: PRJNA1078726 (run SRR28068996).

All the sequences introduced in this study are available at GenBank with the following accession numbers: PP766724 –PP766727, PP727122, PP722687, PP729153, PP725265.

The following information was supplied regarding the registration of a newly described species:

Species name: urn:lsid:zoobank.org:act:3AD06456-46E9-4438-A888-4DAAB0F36A19

Publication LSID: urn:lsid:zoobank.org:pub:49F66ABC-EEAA-4C92-8CCF-6B9200AB91B6

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
