# Peer review of "A new species of alien land flatworm in the Southern United States"

_PeerJ, doi:10.7717/peerj.17904_

## Round 0.1 · original submission · Minor Revisions

Dear Jean-Lou, all the reviewer fing the manuscript appropriate for publication after minor revisions. I recommend you and your coauthors to follow these suggestions as much as possible in your revised version.

Reviewer 1 ·

Basic reporting

Excellent

Experimental design

Excellent

Validity of the findings

Excellent

Additional comments

This is an exemplary manuscript suitable for publication in PeerJ. It describes an interesting and noteworthy case in which only anatomical and molecular data provide sufficient evidence for the identification of specimens of land planarians, in this particular case even representing a new species. Further, it discusses interesting problems associated with the molecular analyses of terrestrial flatworms. I found little at fault regarding the presentation of the text and the figures. In the text of the pdf I have inserted a few Comments regarding suggestions for improvement.

Annotated reviews are not available for download in order to protect the identity of reviewers who chose to remain anonymous.

·

Basic reporting

A new species of Amaga is described by integrative taxonomy in the MS entitled A new species of alien land flatworm in the Southern United States. The entire mitogenome is provided along with phylogenies of some geoplanids inferred from either their mitogenomes or a subset of three molecular markers. The authors provide evidence that the species is new to science and that the species may be indistinguishable from other geoplanid species introduced to the U.S. based on external appearance alone. In the light of hteir data, the authors also make some considerations about what genetic code should be appropriate when dealing with the mitogenome.

The manuscript is clear in its statements and is concise. All figures and tables are necessary. In my opinion, with these minor changes, the MS is suitable for publication in PeerJ (There is no annotated manuscript attached):

• Title: Since the author has made relevant considerations about the genetic code that should be used to translate the protein-coding genes of Geoplanidae, I suggest adding the following to the title: "with comments on the mitochondrial genetic code".
• Line 66-7: “Obama nungara, native to Brazil and Argentina, is now reported in many countries in Europe (Čapka & Čejka 2021; Justine et al. 2024; Justine et al. 2020b; Mori et al. 2022; Soors et al. 2019; Thunnissen et al. 2022),”. Please consider this paper: Lago-Barcia et al. 2018. Reconstructing routes of invasion of Obama nungara (Platyhelminthes: Tricladida) in the Iberian Peninsula. Biol Invasions DOI 10.1007/s10530-018-1834-9
• Line 128: Froehlich 1955. → Froehlich 1954
• Line 187: Geoplanidae Stimpson, 1857 → Geoplanidae Stimpson, 1858 (Please check front cover of the issue (and not the bottom right corner of the unevenly printed pages of Stimpson's paper))
• Line 211: “is broadly lanceolate in shape with tapered extremities” (since lanceolate is a shape)
• Line 218: Section ‘Diagnosis (on preserved specimens)’. This diagnosis is based on preserved and sectioned specimens, but the title leads the reader to find features that can only be observed in preserved specimens. Therefore, I suggest simply stating "diagnosis" and emphasizing "preserved" whenever necessary along the diagnosis.
• Line 276: ‘The parenchymatous mid-pharyngeal wall is comprised of’. I am not familiar with ‘wall’ in this sense. Is it the same as ‘region’?
• Line 239: ‘The prostatic vesicle empties into the male atrium via a progressively enlarged vertical slit in the sulcus’. An introduction to the term ‘sulcus’ is lacking.
• Line 437: ‘The genus Amaga Ogren and Kawakatsu, 1990 (Ogren & Kawakatsu 1990) presently comprises a heterogeneous group of nine eight species’. Currently, A. olivacea (Müller, 1856 in Schultze, 1856) (not 1857) is placed in Pseudogeoplana (see Ogren et al., 1992. Bull. Fuji Women’s College, (30), II: 59-103. Keep this comment in mind whenever P. olivacea is mentioned in the text.
• Lines 507-9: ‘Of passing interest is the pigment surrounding the testes in A. pseudobama. This phenomenon has previously been observed in Amaga expatria (Jones & Sterrer 2005), and in some other Geoplaninae including Obama ladislavii (von Graff, 1899)’. Pigment surrounding testes is also found in Transandiplana graui Almeida et al., 2023 DOI 10.1093/zoolinnean/zlac072/6808355
• The authors might find it useful to mention in the section ‘Conclusion’ that caution should be exercised in annotating the mitochondrial protein-coding genes of Geoplanidae.
• Abbreviations in figures: csd (common sperm duct) and ed (ejaculatory duct) are not mentioend in the description; sv (seminal vesicle) is not descriped in the description section.
• In the References: Fernández-Álvarez FA → Fernández-Álvarez FÁ; Alvarez-Presas → Álvarez-Presas

Fernando Carbayo

Experimental design

No comment

Validity of the findings

No comment.

Additional comments

No comment.

Reviewer 3 ·

Basic reporting

The article entitled: "A new species of alien land flatworm in the Southern United States" is well written and clear. References are correct. Article structure, figures and tables are necessary. Some extra tables are needed, and some information could be added to the existing ones in order to make them completer and more informative. The article is self-contained with relevant results. It is a contribution to the knowledge of invasive terrestrial planarians, describing a new species and sequencing its mitogenome.

The abstract is too long. Please, could the authors make it a little shorter to make it more precise and direct?

Experimental design

No comment.

Validity of the findings

No comment.

Additional comments

I only have some comments/suggestions that I think can be used to try to improve the text.

Material and Methods:
-Detail on authors' contribution is already given in the relevant section of the manuscript, so it is not necessary to detail in M&M who did everything, as all of the mentioned people are listed as coauthors. Skipping so much detail, the text will be more fluent and easier to read and follow.

-Was any filtering applied to the raw reads in order to avoid contaminations?

-How was the 28S of the new species retrieved? This needs to be explained in the M&M section.

-Line 153-156: This information is already given in the previous section. Leave it only in one place.

-It would be useful naming the datasets in order to make clear to which one the authors are referring to in the methods description. Otherwise it is a bit confusing. Same for the results section.

-Why was modeltest used gene by gene in the matrix concatenating 28S, cox1 and EF, and for all the proteins together in the mitochondrial PCG matrix? Is there any particular reason to treat all the proteins together as a whole? Did the authors use partitions in the phylogenetic inference of the mitochondrial PCG? If not, why?

Results:
-Were the 4 vouchers available for sectioning? If so, why not using them as additional material?

-On which specimen is the living specimens' description based?

-I encourage the authors to add a molecular diagnostics section to the results. Given that it is a cryptic species (by external appearance) and that it is very difficult or impossible to diagnose it by external characters (named pictures, for instance), a description of how to describe it based on molecular data will be useful for future diagnoses.

-Molecular information could be added to table 1 in order to be able to identify the specimens easily and gathering all the information together.

-The authors consider that some identifications in iNaturalist were erroneous based on what? Please, specify, and if there are different sources of evidence that give support to the new identification, add it as a new column in Table2.

-Line 417 on: This gene (28S) is duplicated in all the terrestrial planarians, and shows two different types, however, it has been difficult to find one of the two in the sequences present in public databases and in published articles. Please, cite the original paper where the tandem duplication of the ribosomal cluster is described in some Tricladida groups (Carranza et al 1998).

-A table with GenBank accession numbers of the sequences used for the phylogenies is missing.

Discussion:
-Line 493: This is result of using single or a few specimens to describe new species. When possible, it is desirable using at least 3 or more specimens to make the description of the new species, in order to distinguish between species' characters or population variability. A comment on that could be added, also regading the present manuscript where a new species is described based only in a single specimen.

-Line 517: This needs to be tested by other evidence. Please add the indication that this is an assumption.

---

## Round 0.2 · accepted · Accept

All minor revisions have been made to the manuscript. Well done for submitting such a high-quality manuscript.